# Maternal and Dietary Factors Are Associated with Metabolic Syndrome in Women with a Previous History of Gestational Diabetes Mellitus

**DOI:** 10.3390/ijerph192416797

**Published:** 2022-12-14

**Authors:** Farah Yasmin Hasbullah, Barakatun-Nisak Mohd Yusof, Rohana Abdul Ghani, Zulfitri ‘Azuan Mat Daud, Geeta Appannah, Faridah Abas, Sangeetha Shyam

**Affiliations:** 1Department of Dietetics, Faculty of Medicine and Health Sciences, Universiti Putra Malaysia, Serdang 43400, Selangor, Malaysia; 2Diabetes Research Unit, Faculty of Medicine and Health Sciences, Universiti Putra Malaysia, Serdang 43400, Selangor, Malaysia; 3Institute for Social Science Studies, Putra Infoport, Universiti Putra Malaysia, Serdang 43400, Selangor, Malaysia; 4Department of Internal Medicine, Faculty of Medicine, Universiti Teknologi MARA, Sungai Buloh 47000, Selangor, Malaysia; 5Department of Nutrition, Faculty of Medicine and Health Sciences, Universiti Putra Malaysia, Serdang 43400, Selangor, Malaysia; 6Department of Food Science, Faculty of Food Science and Technology, Universiti Putra Malaysia, Serdang 43400, Selangor, Malaysia; 7Unitat de Nutrició Humana, Departament de Bioquímica i Biotecnologia, Universitat Rovira i Virgili, 43201 Reus, Spain; 8Institut d’Investigació Sanitària Pere Virgili (IISPV), Hospital Universitari Sant Joan de Reus, 43204 Reus, Spain; 9Centro de Investigación Biomédica en Red de Fisiopatología de la Obesidad y Nutrición (CIBEROBN), Instituto de Salud Carlos III (ISCIII), 28029 Madrid, Spain; 10Centre for Translational Research, IMU Institute for Research and Development (IRDI), International Medical University (IMU), Kuala Lumpur 57000, Malaysia

**Keywords:** metabolic syndrome, gestational diabetes mellitus, cardiometabolic risk, dietary patterns, Malaysia, low- and middle-income countries, determinants

## Abstract

While it is known that women with a previous history of gestational diabetes mellitus (post-GDM) have a higher risk of metabolic syndrome (MetS), evidence of lifestyle practices from low- and middle-income countries (LMICs) is still scarce. This study aimed to determine the factors associated with MetS in women post-GDM. This cross-sectional study involved 157 women post-GDM (mean age 34.8 ± 5.6 years) sampled from Selangor, Malaysia. We collected data on sociodemographic characteristics and obstetric history. Food intake was assessed using a food frequency questionnaire, and dietary patterns were derived from principal component analysis. MetS was diagnosed according to the 2009 Harmonized criteria. The prevalence of MetS in this study was 22.3%. Western dietary pattern consumption was correlated with MetS, body mass index (BMI), waist circumference, and triglyceride levels. Independent factors associated with MetS were lower education level (odds ratio, OR 4.017, *p* = 0.007), pre-pregnancy BMI (OR 1.192, *p* = 0.002), and Caesarean delivery (OR 3.798, *p* = 0.009). The study identified the maternal and dietary factors associated with MetS in women post-GDM in Malaysia. Community-based interventions that include dietary modification are warranted to prevent MetS and its complications, thus helping to reduce the overall disease burden.

## 1. Introduction

Metabolic syndrome (MetS) is a global public health burden due to its rising prevalence, affecting 10–40% of the world’s population [1]. MetS is a cluster of risk factors consisting of central or abdominal obesity, glucose intolerance or insulin resistance, a pro-thrombotic state, a pro-inflammatory state, and hypertension [2]. Individuals with MetS have a five-fold increased risk of type 2 diabetes (T2D) [3] and a two-fold increased risk of developing cardiovascular diseases (CVD) in the next 5–10 years [4]. This demonstrates an urgent public need to identify individuals with MetS and initiate weight loss interventions via lifestyle or dietary modification to reduce their risks of metabolic diseases [5,6]. The main determinants of MetS appear to be obesity, the consumption of excess energy intake, and a sedentary lifestyle, on top of genetic and metabolic susceptibilities [2,4]. Various diagnostic criteria are available to diagnose MetS [3,4,7,8,9]. However, due to the differences in diagnostic criteria, the prevalence of MetS varies according to the clinical definitions and measurement cut-offs for a specific gender, ethnic group, or population [4]. Thus, the Harmonized criteria attempted to unify the differing definitions of risk factors; the presence of any three components of MetS was adequate to diagnose an individual with MetS [4].

A previous history of gestational diabetes mellitus (GDM) confers a more than three-fold increased risk of developing MetS compared with women with normoglycemic pregnancies [10,11]. Much of the research on women with a previous history of GDM (post-GDM) centered on the obstetric determinants of MetS, including an excessive pre-pregnancy body mass index (BMI) and gestational weight gain [12,13,14]. While the roles of sociodemographic characteristics, socioeconomic status, or lifestyle factors in the development of MetS are poorly documented, there is some evidence linking older maternal age, ethnicity [15], and dietary patterns [16] with the development of T2D in women post-GDM.

Nevertheless, many of the studies on MetS among women post-GDM were conducted in high-income countries [17] such as Italy [18], Finland [12], Canada [19], Hungary [20], and the USA [21]. MetS was estimated to affect 15–25% of the general population in high-income countries [22,23]. The prevalence of MetS in low- and middle-income countries (LMICs) was reported to be even higher, ranging from 11.9% in the Philippines to 49% in Pakistan [24]. Thus, the etiology of MetS may differ between high-income countries, which are typically in Western countries, compared with LMICs including Malaysia. Moreover, compared with Caucasians, South Asians and East Asians have higher visceral fat at any body mass index (BMI) and early β-cell dysfunction [25,26], which may contribute to the development of MetS at an earlier age. A rapid shift in dietary patterns has also been observed in LMICs including Malaysia, with a transition towards modernized traditional diets [27], a more processed or Westernized diet [28], and increasing sugar intake [29,30], which have been associated with cardiometabolic risks [30].

It would be interesting to observe how sociodemographic, obstetric, and dietary factors in LMICs influence the development of MetS. Hence, this study aimed to determine the factors associated with MetS in women post-GDM.

## 2. Materials and Methods

### 2.1. Study Design and Participants

This cross-sectional study involved women post-GDM from the MyNutritype cohort; the methods have been described in detail elsewhere [31]. GDM was diagnosed according to the Malaysian Clinical Practice Guidelines [32]. The study recruited participants from Universiti Putra Malaysia and Seri Kembangan Health Clinic; both sites are located in the state of Selangor, Malaysia. Data collection was conducted from January to April 2021 at Universiti Putra Malaysia, and from November 2021 to February 2022 at Seri Kembangan Health Clinic. During January to February 2021, Malaysia was under the Movement Control Order (MCO) and shifted to the Conditional Movement Control Order (CMCO) during March to April 2021 [33]. Starting from October 2021, more lenient laws were approved, including inter-district traveling, under the National Recovery Plan (NRP) [33]. Data collection was performed in strict accordance with the MCO, CMCO, and NRP laws. However, no data collection was conducted around festival periods, including Eid al-Fitr, Chinese New Year, and Christmas, as well as during Ramadan fasting, in order to avoid significant changes in dietary patterns.

The study included Malaysian women aged 18–49 who were recruited at 6 weeks up to 15 years postpartum. The duration lapse following GDM pregnancies was based on previous local studies conducted in women post-GDM [34,35,36]. Furthermore, a systematic review involving 129 studies found that up to a third of women post-GDM progressed to T2D within 15 years postpartum [37]. The Sister Study, a large prospective study involving 50,884 women post-GDM in the USA also reported that the risk for developing T2D was the greatest within 15 years following GDM pregnancies [38]. We excluded pregnant women; those who were recently hospitalized within the last 6 weeks; and those who had a prior diagnosis of medical conditions such as type 1 or type 2 diabetes, cancer, liver, or renal disorders. The Medical Research Ethics Committee of the Ministry of Health, Malaysia (NMRR-19-3482-50546) and the Research Committee of Universiti Putra Malaysia (JKEUPM-2019-464) approved the study.

Sample size was calculated using the prevalence formula for cross-sectional studies [39]. Based on the 22% prevalence of MetS in Malaysian women post-GDM [40], a 95% confidence level, a 10% tolerable deviation of values around the estimate, and after adjusting for a 20% non-response rate [39], a minimum of 83 participants was required for the study. Participants provided their written informed consent at enrolment.

### 2.2. Diagnosis of Metabolic Syndrome

MetS was diagnosed according to the Joint Interim Statement or Harmonized guidelines [4]. Participants were identified as having MetS if they had 3 or more of the following criteria: (1) central obesity, as indicated by a body mass index (BMI) ≥30 kg/m^2^ or waist circumference ≥80 cm for Asian women; (2) elevated triglycerides (≥1.7 mmol/L); (3) elevated blood pressure (systolic ≥ 130 mmHg and/or diastolic ≥ 85 mmHg); (4) elevated fasting plasma glucose (FPG ≥ 5.6 mmol/L); and (5) reduced HDL-cholesterol (<1.3 mmol/L). Participants were then categorized into MetS (those with metabolic syndrome) or Non-MetS (those without metabolic syndrome) groups.

### 2.3. Sociodemographic Characteristics

Participants’ sociodemographic characteristics were obtained using a structured questionnaire. Data included age, ethnicity (Malay, Chinese, Indian, or other ethnic groups), marital status (married or divorced/widowed), and education level (primary/secondary versus tertiary education). For working status, employed or self-employed participants were considered as working, while housewives and students were categorized as not working. Household income was categorized into 3 groups based on the income range for the Selangor state of Malaysia [41]: low-income (<USD 1490); middle-income (USD 1490–2234); and high-income (≥USD 2235).

### 2.4. Obstetric History

Obstetric data were obtained from participants’ medical records, including gravidity, parity, family history of diabetes, recurrence of GDM, and current breastfeeding status. Index GDM was defined as the most recent GDM pregnancy. Information pertaining to index GDM included duration since index GDM (0–5, 6–10, or 11–15 years post-GDM), pre-pregnancy BMI, gestational age during GDM diagnosis, delivery method (spontaneous vaginal delivery or Caesarean section), treatment (diet control only or diet control with metformin/insulin), breastfeeding practices (never/stopped breastfeeding or still breastfeeding the infant of index GDM) and presence of macrosomia.

### 2.5. Anthropometric, Biochemical and Clinical Measurements

Participants were required to remove outer attire and shoes and empty their pockets for their height and body weight measurement. Height was measured using a stadiometer (SECA model 206, Vogel & Halke GmbH & Co., Hamburg, Germany) while body weight and body fat percentage were measured using a body composition monitor (Tanita Health Equipment Ltd., Tokyo, Japan). BMI was estimated and categorized based on WHO classification [42]. Waist and hip circumferences were measured with a measuring tape (SECA model 203, Vogel & Halke GmbH & Co., Hamburg, Germany), and the waist-to-hip ratio (WHR) was calculated and categorized based on WHO guidelines [43]. A blood pressure monitor (OMRON Corporation, Kyoto, Japan) measured the systolic and diastolic blood pressure with the participants in a seated position. After fasting overnight for 8–12 h, participants’ fasting venous blood samples were taken for FPG, HbA1c, insulin, and lipid profile (total cholesterol, triglycerides, HDL-cholesterol, and LDL-cholesterol) assessments.

### 2.6. Dietary Patterns

Food intake was assessed using a 165-item semi-quantitative food frequency questionnaire (FFQ) adapted from the Malaysian Adult Nutrition Survey [44]. Participants were asked about the types of food and beverages consumed within the past month. The food items were re-categorized into 11 food groups based on similarities in nutrient composition and previous local data to derive the dietary patterns [28,45,46]. The food groups included cereals and grains; fast food; meat, poultry, and eggs; fish and seafood; legumes; milk and dairy products; fruits and vegetables; coffee, tea, and sugar-sweetened beverages; confectionaries; sugar, spreads and creamer; and salty food and condiments.

Principal component analysis (PCA) was performed to obtain the dietary patterns. The Kaiser–Meyer–Olkin value was 0.636, which exceeded the recommended value of 0.50 and indicated that the sample size was adequate to run the PCA [47]. The scree plot and Eigenvalue >1.2 determined the number of dietary patterns extracted [28]. The factor scores were orthogonally rotated by Varimax transformation to increase loading differences for improved interpretability [45,48]. Factor loading scores ≥|0.3| indicated a high or low intake of that food group in dietary patterns [46,48]. Dietary patterns were analyzed only for participants with plausible reporting of energy intake, i.e., 500–3500 kcal/day for women [49].

### 2.7. Statistical Analysis

All statistical tests were performed in SPSS software version 25.0. Categorical data were expressed in numbers and percentages (%), continuous data as mean ± standard deviation (SD), and data normality were assessed prior to data analysis. Characteristics between women with and without MetS (MetS versus Non-MetS groups) were compared using an independent *t*-test or Pearson’s Chi-squared test. Pearson’s correlation coefficient determined the association between dietary patterns and MetS. Binary logistic regression determined the factors significantly associated with MetS. Sociodemographic and obstetric variables significantly associated with MetS at the bivariate level (*p* < 0.05) were entered into a single regression model along with the dietary patterns. Odds ratios (ORs) with 95% confidence intervals (CIs) were presented. The significant value for all statistical tests was set at *p* < 0.05.

## 3. Results

In total, 35 participants (22.3%) were diagnosed with MetS. Among the participants with MetS, elevated waist circumference was the most common MetS component (97.1%), followed by reduced HDL-cholesterol (80.0%) and raised blood pressure (71.4%). Participants with MetS were significantly heavier (*p* < 0.001), had elevated waist and hip circumferences (*p* < 0.001), and had a higher body fat percentage (*p* < 0.001) and systolic and diastolic blood pressure (both *p* < 0.001). Significantly more participants had abdominal obesity in the MetS group, as shown by the WHR (*p* = 0.006). In terms of biochemical profile, the MetS group had significantly higher levels of FPG (*p* = 0.001), fasting insulin (*p* < 0.001), HbA1c (*p* = 0.016), and triglycerides (*p* < 0.001), and lower HDL-cholesterol (*p* < 0.001) (Table 1).

Table 2 compares the sociodemographic and obstetric characteristics of participants with and without MetS. Compared with Non-MetS, the MetS group had a significantly higher pre-pregnancy BMI (*p* = 0.001). GDM was diagnosed earlier in the pregnancy of women with MetS (*p* = 0.036). Significantly fewer people in the MetS group had tertiary education (*p* = 0.043) and were currently breastfeeding (*p* = 0.015). In contrast, there was a higher prevalence of recurrent GDM among participants with MetS (*p* = 0.004), Caesarean delivery (*p* = 0.007), and metformin/insulin use during index GDM (*p* = 0.036) (Table 2).

Seven participants were excluded from the dietary pattern analysis due to dietary misreporting (n = 1 under-reporter with total energy intake <500 kcal/day and n = 6 over-reporters with total energy intake >3500 kcal/day). Two data-derived dietary patterns emerged—the Western and the Prudent dietary patterns, which contributed to 34.0% of the total variances.

The Western dietary pattern was characterized by high intakes of cereals and grains; fast food; coffee, tea, and sugar-sweetened beverages; confectionaries; sugar, spreads, and creamer; and salty food and condiments. The Prudent dietary pattern consisted of high intakes of food groups such as meat, poultry, and eggs; fish and seafood; milk and dairy products; and fruits and vegetables (Figure 1). Appendix A shows the factor loading scores of the identified dietary patterns.

At the bivariate level, the Western dietary pattern was positively and significantly associated with total MetS score, BMI, waist circumference, and triglyceride levels (*p* = 0.043, 0.013, 0.019, and 0.041, respectively) (Table 3).

Multivariate logistic regression analysis showed that the factors significantly associated with MetS were lower education level (OR 4.017, *p* = 0.007), pre-pregnancy BMI (OR 1.192, *p* = 0.002), and Caesarean delivery at index GDM (OR 3.798, *p* = 0.009). The model contributed to 36.8% of the variation in MetS (Table 4).

## 4. Discussion

This study achieved the aim of identifying factors significantly associated with MetS in women post-GDM. Lower education level, pre-pregnancy BMI, and Caesarean delivery were independent factors associated with MetS. The Western dietary pattern was also correlated with MetS, BMI, waist circumference, and triglyceride levels. The prevalence of MetS in women post-GDM in this study was 22.3%, which was comparable with a previous Malaysian study [40] and other LMICs including in India [50] and Sri Lanka [51].

Malaysia is listed among the LMICs by the World Bank [17]. Although the participants in this study were recruited in an urban district, those with MetS had a significantly lower education level. This result aligned with findings from previous local studies [52,53] as well as in high-income countries such as South Korea [54] and the USA [55]. The studies confirmed that a lower education and income were associated with a higher risk of MetS in women [54,55]. The mechanism linking education and MetS may be explained by a few factors, including health-seeking behavior and psychological distress. Those with higher education were less likely to engage in adverse health behaviors such as physical inactivity, smoking, or consuming alcohol, which has been associated with MetS [54,55,56].

A prospective cohort study in the USA found that women with lower education levels also had higher levels of depressive symptoms and lower levels of self-esteem and social support [57]. The association between education and MetS was mediated by low reserve capacity (optimism, self-esteem, and social support) and high negative emotions (depressive symptoms, anger, and tension) [57]. In addition, a higher stress level demonstrated in those with lower socioeconomic status was linked with increased cortisol secretion, which subsequently increased the risk of visceral fat accumulation, contributing to central obesity, one of the components of MetS [53].

Regarding obstetric factors, pre-pregnancy BMI and Caesarean delivery had a significant association with the increased odds of MetS. In Korean women post-GDM, pre-pregnancy obesity has increased the odds of T2D [58]. Meanwhile, in mothers with GDM in Brazil, Caesarean delivery was associated with pre-pregnancy obesity [59], which in turn was associated with MetS [12,13,14]. Maternal obesity has been demonstrated to increase the risk of postpartum diabetes [60], even as early as <1 year postpartum in Korean women post-GDM [61]. Thus, identifying women post-GDM with poor obstetric histories and ensuring normal BMI prior to each conception are crucial strategies that may help to halt the progression of MetS and delay its complications.

The Western dietary pattern was significantly and positively associated with MetS and cardiometabolic risk factors (BMI, waist circumference, and triglyceride levels) in this study. A meta-analysis of 40 observational studies found that the Meat/Western dietary pattern, characterized by high intakes of red and processed meat, animal fat, eggs and sweets, increased the risk of MetS (OR 1.19) [62]. The increased risk of MetS due to Meat/Western dietary pattern consumption was slightly higher in Asian countries compared with European populations (20% vs. 15%) [62]. More studies need to be conducted to investigate why Asian populations, who are mostly from LMICs, appeared to be more prone to this effect. Nevertheless, a possible reason could be due to food insecurity in LMICs, especially in low-income households. Due to increasing global food prices, healthier and more nutrient-dense food such as fresh fish and seafood, dairy and fruits are becoming more expensive [63]. Food insecurity may then result in weight gain and obesity due to the consumption of energy-dense foods which are often cheaper than healthy food [63], leading to the development of MetS.

Similar to previously identified Western dietary patterns [62], our study included processed meat, refined grains, sweets, and sugar-sweetened beverages in the Western dietary pattern. However, this study classified unprocessed meat, poultry, and eggs into the Prudent dietary pattern instead. In addition, this study did not distinguish between low-fat and high-fat dairy products (all dairy was categorized into the Prudent dietary pattern) and refined grains or wholewheat grains (all cereals and grains were categorized into the Western dietary pattern). The classification of food groups was performed mainly according to data from previous local studies [28,45,46]. Data from the nationwide Malaysian Adult Nutrition Survey showed that the Western dietary pattern consumed by the general Malaysian adult population included fast food, bread, bread spreads, sugar-sweetened beverages, confectionaries, and condiments [28], which was reflected in this study. The Western dietary pattern was also positively significantly correlated with BMI, waist circumference, and triglyceride levels, which agreed with findings from the Malmö Diet and Cancer Study involving Swedish adults [64]. The correlation between Western dietary patterns and central obesity may be the most crucial pathway in driving the mechanism behind MetS development and overall cardiometabolic health [64].

The Prudent dietary pattern identified in this study was high in meat, poultry, and eggs; fish and seafood; milk and dairy products; and fruits and vegetables. A review of studies conducted in MetS in several adult populations [62] also identified the Prudent dietary pattern as being high in fruits and vegetables, fish, milk and dairy products, and meat and poultry. However, unlike those studies [62], our study was unable to find a significant correlation between the Prudent dietary pattern and reduced odds of MetS.

The findings from this study have a few implications for public health practice. First, this study identified MetS as early as six weeks postpartum. Hence, community lifestyle interventions that include a dietary modification component are critical to identifying MetS and preventing future development of T2D and CVD. The current literature has recommended cardioprotective dietary patterns that help in preventing MetS and its complications including the Prudent, Mediterranean, and DASH diets [62,65]. Secondly, assessments of socioeconomic background and nutritional status, including dietary intake, have to be routinely conducted during obstetric follow-ups to identify women post-GDM with a high cardiometabolic risk and to reduce the incidence of MetS following their deliveries. Prevention of MetS would benefit LMICs in the long run as they would reduce the disease burden in the community, including treatment costs for T2D and CVD.

The study contributes data on the prevalence and prevention of MetS in women with a previous history of GDM, which is an area with limited evidence, particularly in regard to dietary factors. The Western and Prudent dietary patterns in this study were also slightly different from the diets commonly practiced in Western or high-income countries. This could be due to the culturally unique food frequency questionnaire used [44]; hence, the dietary patterns derived in this study were more representative of the typical Asian diet.

However, this study has a few limitations. This was a cross-sectional study, and as such, no causal relationship can be established. Well-designed prospective cohort studies and randomized controlled trials with adequate sample sizes are recommended to ascertain the findings from our study. Secondly, the study recruited women up to 15 years post-GDM. Hence, some of the women may have progressed to having MetS complications such as T2D or CVD. Future research should take into account the cardiometabolic disease diagnosis and remove the participants from further analysis. Although the study excluded those with a prior diagnosis of type 1 or type 2 diabetes, some of the participants may have had undiagnosed MetS before pregnancy, which is another limitation of the study. Furthermore, we did not study cardiometabolic risk biomarkers, such as hs-CRP, which were shown to be elevated in adults and women post-GDM with MetS [18,21,66,67], or adiponectin, which was reduced in MetS [67,68]. Thus, the assessment of cardiometabolic risk biomarkers should be performed in future studies to better understand the physiology behind the development of MetS. Lastly, data collection took place during various phases of the MCO following the COVID-19 outbreak. There may have been changes in dietary patterns due to the MCO during the COVID-19 pandemic, as shown in the MyNutriLifeCOVID-19 study [69]; however, dietary changes were not assessed in this study.

## 5. Conclusions

The study identified the maternal and dietary factors associated with MetS in women post-GDM in a LMIC country. The consumption of Western dietary patterns was correlated with BMI, waist circumference, triglyceride levels, and overall MetS score. Independent factors associated with the increased odds of MetS included lower education level, pre-pregnancy BMI, and Caesarean delivery. The findings highlight the importance of assessing sociodemographic background, obstetric history, and nutritional status to identify women post-GDM at increased risk of MetS. Community-based interventions that include dietary modifications are warranted to prevent MetS and its complications, thus helping to reduce the overall disease burden.

## Figures and Tables

**Figure 1 ijerph-19-16797-f001:**
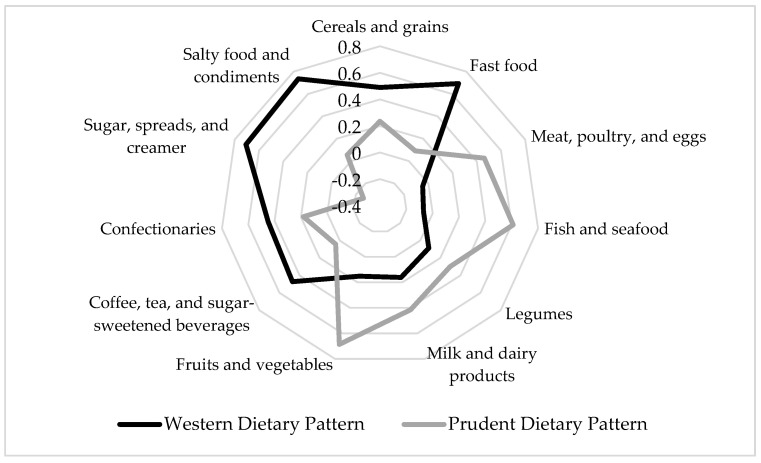
Diagram of factor loading scores of dietary patterns tested using principal component analysis. Only food groups with factor loading scores ≥|0.3| are presented. The Western dietary pattern has high intakes of cereals and grains; fast food; coffee, tea, and sugar-sweetened beverages; confectionaries; sugar, spreads, and creamer; and salty food and condiments. The Prudent dietary pattern has high intakes of meat, poultry, and eggs; fish and seafood; milk and dairy products; and fruits and vegetables.

**Table 1 ijerph-19-16797-t001:** Metabolic syndrome and cardiometabolic risk factors in participants (*n* = 157).

MetS Criteria	MetS (*n* = 35)	Non-MetS (*n* = 122)	*p*-Value
Body mass index ≥30 kg/m^2^	12 (34.3)	23 (18.9)	0.053
Central obesity (waist circumference ≥80 cm)	34 (97.1)	82 (67.2)	<0.001 *
Raised triglycerides (≥1.7 mmol/L)	19 (54.3)	2 (1.6)	<0.001 *
Reduced HDL-cholesterol (<1.3 mmol/L)	28 (80.0)	23 (18.9)	<0.001 *
Raised blood pressure (systolic ≥130 mmHg and/or diastolic ≥85 mmHg)	25 (71.4)	25 (20.5)	<0.001 *
Raised fasting plasma glucose (≥5.6 mmol/L)	15 (42.9)	0 (0.0)	0.002 *
Anthropometric and clinical measurements			
Height (m)	1.58 ± 0.05	1.56 ± 0.06	0.198
Weight (kg)	74.2 ± 13.9	62.7 ± 12.5	<0.001 *
Body mass index (kg/m^2^)	29.8 ± 5.2	25.6 ± 4.5	<0.001 *
Underweight/normal	5 (14.3)	62 (50.8)	
Overweight/obese	30 (85.7)	60 (49.2)	<0.001 *
Waist circumference (cm)	93.8 ± 10.0	85.2 ± 10.3	<0.001 *
Hip circumference (cm)	111.7 ± 10.7	104.4 ± 9.5	<0.001 *
Waist-to-hip ratio	0.84 ± 0.05	0.83 ± 0.13	0.547
Within recommendation (≤0.8)	6 (2.9)	52 (42.6)	
Abdominal obesity (>0.8)	29 (82.9)	70 (57.4)	0.006 *
Body fat (%)	43.7 ± 5.5	38.3 ± 6.4	<0.001 *
Systolic blood pressure (mmHg)	120 ± 17	109 ± 14	<0.001 *
Diastolic blood pressure	87 ± 10	78 ± 10	<0.001 *
Biochemical profiles			
Fasting plasma glucose (mmol/L)	5.79 ± 2.01	4.55 ± 0.45	0.001 *
Fasting insulin (uIU/mL)	14.4 ± 9.3	6.1 ± 4.4	<0.001 *
HbA1c (%)	6.1 ± 1.5	5.4 ± 0.3	0.016 *
Total cholesterol (mmol/L)	5.28 ± 1.00	5.27 ± 0.88	0.96
Triglycerides (mmol/L)	1.91 ± 0.88	0.92 ± 0.35	<0.001 *
HDL-cholesterol (mmol/L)	1.21 ± 0.29	1.63 ± 0.40	<0.001 *
LDL-cholesterol (mmol/L)	3.19 ± 0.89	3.22 ± 0.83	0.877

Values for continuous variables are expressed in mean ± SD and analyzed using an independent *t*-test. Values for categorical variables are expressed in number (%) and analyzed using Pearson’s Chi-squared tests. * *p* < 0.05.

**Table 2 ijerph-19-16797-t002:** Comparison of sociodemographic and obstetric characteristics between participants with and without metabolic syndrome (*n* = 157).

Variable	MetS (*n* = 35)	Non-MetS (*n* = 122)	*p*-Value
	Mean ± SD or *n* (%)		
Sociodemographic characteristics			
Age (years)	34.8 ± 5.8	34.8 ± 5.6	0.995
Ethnicity			
Malay	31 (88.6)	100 (82.0)	
Chinese	1 (2.9)	12 (9.8)	0.395
Indian	3 (8.6)	6 (4.9)	
Others	0 (0.0)	4 (3.3)	
Marital status			0.999
Married	35 (100.0)	120 (98.4)	
Divorced/widowed	0 (0.0)	2 (1.6)	
Education level			
Primary/secondary education	22 (62.9)	53 (43.4)	0.043 *
Tertiary education	13 (37.1)	69 (56.6)	
Working status			
Working	24 (68.6)	83 (68.0)	0.999
Not working	11 (31.4)	39 (32.0)	
Household income (USD) ^1^	1256 ± 771	1366 ± 916	
Low-income group	23 (65.7)	73 (59.8)	0.520
Middle-income group	11 (31.4)	40 (32.8)	
High-income group	1 (2.9)	9 (7.4)	0.594
Household size	5 ± 2	5 ± 1	0.687
Obstetric history			
Family history of diabetes	28 (80.0)	85 (69.7)	0.230
Gravidity	3.3 ± 2.2	3.1 ± 1.9	0.569
Parity	2.6 ± 1.5	2.6 ± 1.4	0.969
Recurrent GDM	12 (34.3)	22 (18.0)	0.040 *
Currently breastfeeding	15 (42.9)	80 (65.6)	0.015 *
Duration since index GDM (years)	2.7 ± 3.9	2.5 ± 3.4	
0–5 years	29 (82.9)	99 (81.1)	0.802
6–10 years	3 (8.6)	19 (15.6)	
11–15 years	3 (8.6)	4 (3.3)	0.277
Pre-pregnancy BMI (kg/m^2^)	27.4 ± 5.4	23.8 ± 3.9	0.001 *
Gestational age during index GDM diagnosis (weeks)	17.3 ± 6.6	20.3 ± 7.7	0.036 *
Delivery method of index GDM			
Spontaneous vaginal delivery	15 (42.9)	86 (70.5)	0.007 *
Caesarean section	18 (51.4)	36 (29.5)	
Treatment during index GDM			
Diet control only	26 (74.3)	108 (88.5)	0.036 *
Diet control with metformin/insulin	9 (25.7)	14 (11.5)	
Breastfeeding status of infant of index GDM			
Never/stopped breastfeeding	19 (54.3)	44 (36.1)	0.077
Still breastfeeding	16 (45.7)	78 (63.9)	
Breastfeeding duration of infant of index GDM (months)	11.3 ± 13.2	10.5 ± 11.2	0.699
Macrosomic infant of index GDM	2 (5.7)	5 (4.1)	0.641

Values for continuous variables are expressed in mean ± SD and analyzed using an independent *t*-test. Values for categorical variables are expressed in number (%) and analyzed using Pearson’s Chi-squared tests. * *p* < 0.05. ^1^ Household income classification was specific to the Selangor population in Malaysia [41].

**Table 3 ijerph-19-16797-t003:** Correlations of dietary patterns with MetS and cardiometabolic risk factors.

	MetS Total Score	BMI	WC	TAG	HDL	SBP	DBP	FPG
Western dietary patterns								
R	0.165	0.202	0.191	0.167	−0.145	−0.013	0.034	0.138
*p*	0.043 *	0.013 *	0.019 *	0.041 *	0.077	0.873	0.677	0.092
Prudent dietary patterns								
R	−0.151	−0.12	−0.153	−0.117	0.100	−0.053	−0.088	−0.065
*p*	0.066	0.145	0.061	0.153	0.221	0.518	0.286	0.428

BMI—body mass index; DBP—diastolic blood pressure; FPG—fasting plasma glucose; HDL—HDL-cholesterol; MetS—metabolic syndrome; SBP—systolic blood pressure; TAG—triglycerides; WC—waist circumference; R—Pearson’s correlation coefficient, and * *p* < 0.05.

**Table 4 ijerph-19-16797-t004:** Odds ratio and 95% confidence intervals for factors associated with metabolic syndrome.

Variables	OR	95% CIs	*p*-Value
Lower education level	4.017	1.474, 10.945	0.007 *
Recurrent GDM	1.857	0.573, 6.016	0.302
Currently breastfeeding	0.384	0.138, 1.067	0.066
Pre-pregnancy body mass index	1.192	1.067, 1.332	0.002 *
Gestational age at index GDM diagnosis	0.961	0.896, 1.031	0.271
Caesarean delivery at index GDM	3.798	1.386, 10.407	0.009 *
Metformin/insulin therapy during index GDM	1.482	0.341, 6.438	0.599
Western dietary pattern	1.337	0.827, 2.163	0.236
Prudent dietary pattern	0.956	0.597, 1.531	0.853

CI—confidence interval; GDM—gestational diabetes mellitus; OR—odds ratio; and * *p* < 0.05 analyzed using binary logistic regression.

## Data Availability

Not applicable.

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
