# Peer review of "Maternal and Dietary Factors Are Associated with Metabolic Syndrome in Women with a Previous History of Gestational Diabetes Mellitus"

_ijerph, 2022, doi:10.3390/ijerph192416797_

Round 1
Reviewer 1 Report
Please complete the chapter Material and method with information regarding the period in which the study took place.
Food patterns can differ depending on the season, that s why it is important to see if the results obtained are averages of the entire calendar year or of specific periods of the year.
line 86 "This cross-sectional study involved women post-GDM from the MyNutritype cohort;" - Provide some information regarding the period in which the study was conducted.
Author Response
Comment 1: Please complete the chapter, Material and method with information regarding the period in which the study took place.
Food patterns can differ depending on the season, that’s why it is important to see if the results obtained are averages of the entire calendar year or of specific periods of the year.
line 86 "This cross-sectional study involved women post-GDM from the MyNutritype cohort;" - Provide some information regarding the period in which the study was conducted.
Response: The study period was added in 2.1 Study Design and Participants (page 2-3)
“Data collection was conducted in January to April 2021 at Universiti Putra Malaysia, and November 2021 to February 2022 at Seri Kembangan Health Clinic. During January to February 2021, Malaysia was under the Movement Control Order (MCO) and shifted to the Conditional Movement Control Order (CMCO) during March to April 2021 [33]. Starting from October 2021, more lenient laws were approved, including inter-district traveling, under the National Recovery Plan (NRP) [33]. Data collection was performed in strict accordance with the MCO, CMCO and NRP laws. However, no data collection was conducted around festival periods, including Eid al-Fitr, Chinese New Year and Christmas as well as during Ramadan fasting to avoid significant changes in dietary patterns.”
Reference
[33] Malaysian National Security Council. Available online: https://www.mkn.gov.my (accessed on 5 December 2022).
Added in 2.6 Dietary Patterns (page 4)
“Participants were asked about the types of food and beverages consumed within the past month.”
As a side note, the recruitment was conducted in these two separate periods due to the Movement Control Order restriction laws in Malaysia. Hence, since the food frequency questionnaire asks about participants’ food intake within the past month from the day of the visit, the food intake was from specific periods of the year: from December 2020 to March 2021, then October 2021 to January 2022.
Malaysia is a tropical country and food intake is quite stable throughout the year. For instance, some fruits and vegetables may vary according to the season. Nonetheless, fruits and vegetables are heavily imported due to the high demand, and thus are readily available for consumption (Shahar et al., 2021).
Reference: Shahar, S.; Shahril, M.R.; Abdullah, N.; Borhanuddin, B.; Kamaruddin, M.A.; Yusuf, N.A.M.; Dauni, A.; Rosli, H.; Zainuddin, N.S.; Jamal, R. Development and Relative Validity of a Semiquantitative Food Frequency Questionnaire to Estimate Dietary Intake among a Multi-Ethnic Population in the Malaysian Cohort Project. Nutrients 2021, 13, 1163. https://doi.org/10.3390/nu13041163
Malaysia is also a multicultural country, and there are several festivals throughout the year: Chinese New Year (February), Thaipusam (January), Deepavali (November), and Christmas (December). Although there are some unique festival foods (such as mooncakes for Chinese New Year), these festivals are only celebrated for 1-3 days and are public holidays, hence no data collection was conducted around festival periods. Since Malaysia is a Muslim-majority country, we also did not conduct data collection during Ramadan and Eid al-Fitr (April-May 2021) to avoid significant changes in dietary patterns. Therefore, no correction of intake was made for variation during festival period.
Also added another limitation in the Discussion section (page 10)
“Lastly, data collection took place during various phases of the MCO following the COVID-19 outbreak. There may be changes in dietary patterns due to the MCO during the COVID-19 pandemic, as shown in the MyNutriLifeCOVID-19 study [69]; however, dietary changes were not assessed in this study.”
Reference:
[69] Chin, Y. S.; Woon, F. C.; Chan, Y. M. The Impact of Movement Control Order During the COVID-19 Pandemic on Lifestyle Behaviours and Body Weight Changes: Findings from the MyNutriLifeCOVID-19 Online Survey. PLoS One 2022, 17 (1): e0262332. https://doi.org/10.1371/journal.pone.0262332
Reviewer 2 Report
Abstract:
§ In addition to putting the mean age, it is also interesting to add the standard deviation to give the reader a greater breadth of understanding about the characterization of the included sample.
§ At the end of the abstract it says: “The study identified the maternal and dietary factors associated with MetS in women post-GDM in a LMIC”. However, as already mentioned in the summary itself, this type of assessment is scarce in LMIC countries. With that, the conclusion of this work should be extrapolated only to the study country and not generalize, as was done in the sentence above.
§ In the keywords, terms such as metabolic syndrome were used; gestational diabetes mellitus, already described in the title of the work. It is interesting to replace them with similar terms in order to increase the scope of works on scientific search platforms.
Introduction:
§ In lines 51 and 52, it says: “demonstrating an urgent public need to identify individuals with MetS to prevent the onset of metabolic diseases”. However, obesity is one of the criteria for classifying metabolic syndrome. You need to review this information.
Methodology:
§ Are Malaysian Clinical Practice Guidelines for diabetes different from globally accepted recommendations? If not, could you cite both references, and thus facilitate the extrapolation of the results of this study to other countries.
§ What was the criteria used to include women 6 weeks to 15 years post gestational diabetes? Did you take any reference as a basis? Need to detail in the text.
§ Regarding Sociodemographic Characteristics, it is necessary to describe in the methodology how the classification categories were, as well as what they did for income. The same should be done for obstetric and biochemical data. Based on what references were they classified? Quote in text.
§ The choice of statistical tests was adequate.
Results and discussion:
§ Even though it has been described in the methodology, it is important to put the statistical tests used to obtain the exposed data at the bottom of the tables.
§ What was considered prudent dietary pattern? Did you use any references to classify them that way? It needs to be described in the methodology and also be further explored in the discussion. It is quoted briefly in line 259.
§ How to know if the women included did not already have undiagnosed metabolic syndrome before pregnancy? I believe this is a limitation of the study.
§ I missed justifying why it would be important to carry out this study specifically with women after gestational diabetes. What motivated you to use this audience?
§ Figure 1 needs a short caption, explaining the image.
Conclusion:
§ It needs to be more objective and say briefly if the objective of the study was achieved and what were the main findings.
Author Response
ABSTRACT
Comment 1: In addition to putting the mean age, it is also interesting to add the standard deviation to give the reader a greater breadth of understanding about the characterization of the included sample.
Response: Added in Abstract (page 1)
“This cross-sectional study involved 157 women post-GDM (mean age 34.8 ± 5.6 years) sampled from Selangor, Malaysia.”
Comment 2: At the end of the abstract it says: “The study identified the maternal and dietary factors associated with MetS in women post-GDM in a LMIC”. However, as already mentioned in the summary itself, this type of assessment is scarce in LMIC countries. With that, the conclusion of this work should be extrapolated only to the study country and not generalize, as was done in the sentence above.
Response: Amended Abstract (page 1)
“The study identified the maternal and dietary factors associated with MetS in women post-GDM in Malaysia.”
Comment 3: In the keywords, terms such as metabolic syndrome were used; gestational diabetes mellitus, already described in the title of the work. It is interesting to replace them with similar terms in order to increase the scope of works on scientific search platforms.
Response: Added three other keywords (page 1)
“...Malaysia; low and middle-income countries; determinants”
INTRODUCTION
Comment 4: In lines 51 and 52, it says: “demonstrating an urgent public need to identify individuals with MetS to prevent the onset of metabolic diseases”. However, obesity is one of the criteria for classifying metabolic syndrome. You need to review this information.
Response:
Obesity is one of the criteria for classifying MetS and thus achieving weight loss is a mainstay of therapy for T2DM prevention. Therefore, understanding the role of lifestyle modifications including a dietary pattern is critical to induce significant weight loss. For example, lifestyle intervention (7% weight loss and 150 minutes of physical activity per week) reduced the risk of T2D by 58% in the Diabetes Prevention Program Study [5]. An umbrella review also reported that healthy dietary patterns such as the Mediterranean or DASH diet, or combined dietary and exercise interventions, significantly decreased the risk of T2D [6].
References:
[5] Diabetes Prevention Program Research Group. Reduction in the Incidence of Type 2 Diabetes with Lifestyle Intervention or Metformin. N Engl J Med 2002, 346 (6), 393-403. https://doi.org/10.1056%2FNEJMoa012512
[6] Toi, P.L.; Anothaisintawee, T.; Chaikledkaew, U.; Briones, J.R.; Reutrakul, S.; Thakkinstian, A. Preventive Role of Diet Interventions and Dietary Factors in Type 2 Diabetes Mellitus: An Umbrella Review. Nutrients 2020, 12, 2722. https://doi.org/10.3390/nu12092722
Hence, we have amended Section 1 Introduction (page 2)
“This demonstrates an urgent public need to identify individuals with MetS and initiate weight loss interventions via lifestyle or dietary modification to reduce their risks of metabolic diseases [5,6].”
METHODOLOGY
Comment 5: Are Malaysian Clinical Practice Guidelines for diabetes different from globally accepted recommendations? If not, could you cite both references, and thus facilitate the extrapolation of the results of this study to other countries.
Response: There is no global consensus on the diagnostic thresholds of GDM. However, the Malaysian CPG differs from other guidelines (refer table below).
Table: Selected published criteria for diagnosis of GDM
|
Organization (Year of publication) |
OGTT glucose load |
Plasma glucose concentration thresholds (mmol/L) |
|||
|
Fasting |
1-hour |
2-hour |
3-hour |
||
|
NDDG (1979)1 |
100g |
5.8 |
10.5 |
9.1 |
8.0 |
|
CC (1982)1 |
100g |
5.3 |
- |
8.6 |
7.8 |
|
WHO (1999)2 |
75g |
7.0 |
- |
11.1 |
- |
|
ADA (2010)1 |
75g |
5.3 |
10.0 |
8.6 |
- |
|
100g |
5.3 |
10.0 |
8.6 |
7.8 |
|
|
IADPSG (2010)2 |
75g |
5.1 |
10.0 |
8.5 |
- |
|
WHO (2013) |
75g |
5.1-6.9 |
10.0 |
8.5-11.0 |
- |
|
NICE (2015)2 |
75g |
5.6 |
- |
7.8 |
- |
|
MOH Malaysia (2017)2 |
75g |
5.1 |
- |
7.8 |
- |
NDDG: National Diabetes Data Group, CC: Carpenter & Coustan , WHO: World Health Organization, ADA: American Diabetes Association, IADPSG: International Association of Diabetes and Pregnancy Study Groups, NICE: National Institute for Health and Care Excellence
1GDM diagnosed if ≥2 abnormal value
2GDM diagnosed if ≥1 abnormal value
Nonetheless, findings from several studies documented that a difference in diagnostic criteria affected the prevalence of GDM, but not adverse maternal outcomes or the prevalence of T2D after delivery (Tehrani et al., 2021; Noctor & Dunne, 2015). Regardless of the diagnostic criteria, a previous history of GDM still confers a high risk of developing T2D in later life, and the risk factors predicting T2D remain similar across different diagnostic criteria (Noctor & Dunne, 2015).
References:
Ramezani Tehrani, F.; Naz, M. S. G.; Yarandi, R. B.; Behboudi-Gandevani, S. The Impact of Diagnostic Criteria for Gestational Diabetes Mellitus on Adverse Maternal Outcomes: A Systematic Review and Meta-Analysis. J Clin Med 2021, 10(4), 666. https://doi.org/10.3390/jcm10040666
Noctor, E.; Dunne, F. P. Type 2 diabetes after gestational diabetes: The influence of changing diagnostic criteria. World J Diabetes 2015, 6(2), 234–244. https://doi.org/10.4239/wjd.v6.i2.234
Comment 6: What was the criteria used to include women 6 weeks to 15 years post gestational diabetes? Did you take any reference as a basis? Need to detail in the text.
Response: Amended 2.1 Study Design and Participants (page 3)
“The study included Malaysian women aged 18-49 who were recruited at 6 weeks up to 15 years postpartum. The duration lapse following GDM pregnancies was based on previous local studies conducted in women post-GDM [34-36]. Furthermore, a systematic review involving 129 studies found that up to a third of women post-GDM progressed to T2D within 15 years postpartum [37]. The Sister Study, a large prospective study involving 50,884 women post-GDM in the USA also reported the risk for developing T2D was the greatest within 15 years following GDM pregnancies [38].
References:
[34] Chew, W. F.; Rokiah, P.; Chan, S. P.; Chee, W. S. S.; Lee, L. F.; Chan, Y. M. Prevalence of Glucose Intolerance,aAnd Associated Antenatal and Historical Risk Factors Among Malaysian Women with a History of Gestational Diabetes Mellitus. Singapore Med J 2012, 53(12), 814–820.
[35] Fatin, A.; Alina, T. I. Proportion of Women with History of Gestational Diabetes Mellitus Who Performed an Oral Glucose Test at Six Weeks Postpartum in Johor Bahru with Abnormal Glucose Tolerance. Malays Fam Physician 2019, 14(3), 2–9.
[36] Logakodie, S.; Azahadi, O.; Fuziah, P.; Norizzati, B. I. B.; Tan, S. F.; Zienna, Z. Z. R.; Norliza, M.; Noraini, J.; Hazlin, M.; Noraliza, M. Z.; Sazidah, M. K.; Mimi, O. Gestational Diabetes Mellitus: The Prevalence, Associated Factors and Foeto-Maternal Outcome of Women Attending Antenatal Care. Malays Fam Physician 2017, 12(2), 9–17.
[37] Dennison, R. A.; Chen, E. S.; Green, M. E.; Legard, C.; Kotecha, D.; Farmer, G.; Sharp, S. J.; Ward, R. J.; Usher-Smith, J. A.; Griffin, S. J. The Absolute and Relative Risk of Type 2 Diabetes After Gestational Diabetes: A Systematic Review and Meta-Analysis of 129 Studies. Diabetes Res Clin Pract. 2021, 171, 108625. https://doi.org/10.1016/j.diabres.2020.108625
[38] Diaz-Santana, M. V.; O'Brien, K. M.; Park, Y. M.; Sandler, D. P.; Weinberg, C. R. Persistence of Risk for Type 2 Diabetes After Gestational Diabetes Mellitus. Diabetes Care 2022, 45(4), 864–870. https://doi.org/10.2337/dc21-1430
Comment 7: Regarding Sociodemographic Characteristics, it is necessary to describe in the methodology how the classification categories were, as well as what they did for income. The same should be done for obstetric and biochemical data. Based on what references were they classified? Quote in text.
Response: Amended 2.3 Sociodemographic Characteristics (page 3)
“Participants' sociodemographic characteristics were obtained using a structured questionnaire. Data included age, ethnicity (Malay, Chinese, Indian or other ethnic groups), marital status (married or divorced/widowed), and education level (primary/secondary versus tertiary education). For working status, employed or self-employed participants were considered as working, while housewives and students were categorized as not working. Household income was categorized into 3 groups based on the income range for the Selangor population in Malaysia [41]: low-income (<USD 1490); middle-income (USD 1490–2234); and high-income (≥USD 2235).”
Amended 2.4 Obstetric History (page 3)
“Obstetric data were obtained from participants' medical records, including gravidity, parity, family history of diabetes, recurrence of GDM, and current breastfeeding status. Index GDM was defined as the most recent GDM pregnancy. Information pertaining to index GDM included duration since index GDM (0-5, 6-10, or 11-15 years post-GDM), pre-pregnancy BMI, gestational age during GDM diagnosis, delivery method (spontaneous vaginal delivery or Caesarean section), treatment (diet control only or diet control with metformin/insulin), breastfeeding practices (never/stopped breastfeeding or still breastfeeding infant of index GDM) and presence of macrosomia.”
Amended 2.5 Anthropometric, Biochemical, and Clinical Measurement (page 4)
“…After fasting overnight for 8 – 12h, participants' fasting venous blood samples were taken for FPG, HbA1c, insulin, and lipid profile (total cholesterol, triglycerides, HDL-cholesterol, and LDL-cholesterol) assessments.”
Comment 8: The choice of statistical tests was adequate.
Response: Thank you for the comment.
RESULTS AND DISCUSSION
Comment 9: Even though it has been described in the methodology, it is important to put the statistical tests used to obtain the exposed data at the bottom of the tables.
Response: Added footnote under Table 1 (page 5) and Table 2 (page 7)
“Values for continuous variables are expressed in mean ± SD and analyzed using independent t-test. Values for categorical variables are expressed in number (%) and analyzed using Pearson’s Chi-squared tests. *p < 0.05.”
Amended caption for Figure 1 (page 7)
“Figure 1. Diagram of factor loading scores of dietary patterns tested using principal component analysis”
Table 3 footnote already mentioned “R: Pearson’s correlation coefficient, *p < 0.05.”
Amended footnote under Table 4 (page 8)
“…*p < 0.05 analyzed using binary logistic regression”
Comment 10: What was considered prudent dietary pattern? Did you use any references to classify them that way? It needs to be described in the methodology and also be further explored in the discussion. It is quoted briefly in line 259.
Response: The characteristics of the Western vs Prudent dietary pattern are not mentioned in the Methodology. This is because the dietary patterns derived from PCA are a posteriori (data-derived), and not a priori (hypothesis-derived). A posteriori dietary patterns are not decided in advance and are based on dietary data. On the contrary, a priori diets such as the Mediterranean or DASH diet already have pre-defined characteristics.
Reference: Ocké, M. Evaluation of Methodologies for Assessing the Overall Diet: Dietary Quality Scores and Dietary Pattern Analysis. Proc Nutr Soc 2013, 72(2), 191-199. https://doi:10.1017/S0029665113000013
Thus, we have elaborated on the characteristics of the Prudent dietary pattern in the Discussion section instead (page 9).
“The Prudent dietary pattern identified in this study was high in meat, poultry and eggs, fish and seafood, milk and dairy products, fruits and vegetables. A review of studies conducted in MetS in several adult populations [62] also identified the Prudent dietary pattern as being high in fruits and vegetables, fish, milk and dairy products, and meat and poultry. However, unlike those studies [62], our study was unable to find a significant correlation between the Prudent dietary pattern and reduced odds of MetS.”
Comment 11: How to know if the women included did not already have undiagnosed metabolic syndrome before pregnancy? I believe this is a limitation of the study.
Response: We mentioned in 2.1 Study Design and Participants (page 3) that we excluded those who had a prior diagnosis of medical conditions such as type 1 or type 2 diabetes, cancer, liver, or renal disorders.
However, we agreed that perhaps some of the women included may already have undiagnosed MetS before pregnancy, particularly those individuals with obesity or had dyslipidemia. Thus, we have added in Discussion (page 10)
“Although the study excluded those with a prior diagnosis of type 1 or type 2 diabetes, some of the participants may have had undiagnosed MetS before pregnancy, which is another limitation of the study.”
Comment 12: I missed justifying why it would be important to carry out this study specifically with women after gestational diabetes. What motivated you to use this audience?
Response: Our rationale for conducting study on MetS in women post-GDM has been mentioned in Introduction (page 2). This is because women post-GDM has a high risk of developing MetS and MetS promotes future metabolic diseases including T2D and CVD.
“A previous history of gestational diabetes mellitus (GDM) confers more than three-fold increased risk of developing MetS compared to women with normoglycaemic pregnancies [10,11].”
However, there is limited evidence in sociodemographic or lifestyle factors associated with MetS in women post-GDM, as many of the studies tended to focus on obstetric history.
“Much of the research on women with a previous history of GDM (post-GDM) centered on the obstetric determinants of MetS, including excessive pre-pregnancy body mass index (BMI) and gestational weight gain [12-14]. While the roles of sociodemographic characteristics, socioeconomic status, or lifestyle factors in the development of MetS are poorly documented, there is some evidence linking older maternal age, ethnicity [15] and dietary patterns [16] with the development of T2D in women post-GDM.”
We also added the importance of studying MetS in women post-GDM specifically in a low- and middle-income country.
“Nevertheless, many of the studies on MetS among women post-GDM were conducted in high-income countries [17] such as Italy [18], Finland [12], Canada [19], Hungary [20] and the USA [21]. MetS was estimated to affect 15-25% of the general population in high-income countries [22,23]. The prevalence of MetS in low- and middle-income countries (LMICs) was reported to be even higher, ranging from 11.9% in the Philippines to 49% in Pakistan [24]. Thus, the etiology of MetS may differ between high-income countries, which are typically in Western countries, compared to LMICs including Malaysia. Moreover, compared to Caucasians, South Asians and East Asians have higher visceral fat at any body mass index (BMI) and early β-cell dysfunction [25,26], which may contribute to the development of MetS at an earlier age.”
Comment 13: Figure 1 needs a short caption, explaining the image.
Response: Added caption under Figure 1 (page 7)
“Only food groups with factor loading scores ≥|0.3| are presented. The Western dietary pattern has high intakes of cereals and grains; fast food; coffee, tea, and sugar-sweetened beverages; confectionaries; sugar, spreads, and creamer; and salty food and condiments. The Prudent dietary pattern has high intakes of meat, poultry and eggs; fish and seafood; milk and dairy products; and fruits and vegetables.”
CONCLUSION
Comment 14: It needs to be more objective and say briefly if the objective of the study was achieved and what were the main findings.
Response: Amended the main findings and stated that the objective of the study was achieved in Discussion (page 8)
“This study achieved the aim of identifying factors significantly associated with MetS in women post-GDM. Lower education level, pre-pregnancy BMI, and Caesarean delivery were independent factors associated with MetS. The Western dietary pattern was also correlated with MetS, BMI, waist circumference, and triglycerides level. The prevalence of MetS in women post-GDM in this study was 22.3%, which was comparable with a previous Malaysian study [40] and other LMICs including in India [50] and Sri Lanka [51].”
Reviewer 3 Report
Title: Maternal and Dietary Factors are Associated with Metabolic Syndrome in Women with a Previous History of Gestational Diabetes Mellitus
Manuscript ID: ijerph-2040148
Overall, this study by Hasbullah et al. has a fine rationale since a higher risk of Mets in women with gestational diabetes has been reported in previous studies. The authors have shown the significant role of diet, as a modifiable factor, in the developing of MetS. Besides, they highlighted that attention should be given to preventing or delaying the onset of MetS in GDM mothers. Yet, I have a few comments regarding the manuscript:
· In agreement with the authors, I believe that the cross-sectional design of the study wouldn’t make a cause-effect relationship. Still, the result is worthy to note due to dramatic lifestyle changes, in particular a shift toward westernized diet, in all populations.
· Line 289: Authors say that “Furthermore, we did not study cardiometabolic risk biomarkers such as adiponectin or hs-CRP, which were shown to be elevated in adults with MetS”. However, Adiponectin levels are decreased in adults with MetS. Studies have shown that plasma adiponectin concentrations correlate negatively with the MetS components including waist circumference, visceral fat area, serum triglyceride concentration, fasting plasma glucose, fasting plasma insulin, and systolic and diastolic blood pressure [1]. Besides, a positive correlation was found between plasma adiponectin and high-density lipoprotein cholesterol concentrations [1].
Reference
Ryo M, Nakamura T, Kihara S, Kumada M, Shibazaki S, Takahashi M, Nagai M, Matsuzawa Y, Funahashi T. Adiponectin as a biomarker of the metabolic syndrome. Circulation journal. 2004;68(11):975-81.
Author Response
General comment: Overall, this study by Hasbullah et al. has a fine rationale since a higher risk of Mets in women with gestational diabetes has been reported in previous studies. The authors have shown the significant role of diet, as a modifiable factor, in the developing of MetS. Besides, they highlighted that attention should be given to preventing or delaying the onset of MetS in GDM mothers. Yet, I have a few comments regarding the manuscript:
In agreement with the authors, I believe that the cross-sectional design of the study wouldn’t make a cause-effect relationship. Still, the result is worthy to note due to dramatic lifestyle changes, in particular a shift toward westernized diet, in all populations.
Response: We thank you for your feedback. References have been re-numbered due to revisions made in the manuscript.
Comment 1: Line 289: Authors say that “Furthermore, we did not study cardiometabolic risk biomarkers such as adiponectin or hs-CRP, which were shown to be elevated in adults with MetS”. However, Adiponectin levels are decreased in adults with MetS. Studies have shown that plasma adiponectin concentrations correlate negatively with the MetS components including waist circumference, visceral fat area, serum triglyceride concentration, fasting plasma glucose, fasting plasma insulin, and systolic and diastolic blood pressure [1]. Besides, a positive correlation was found between plasma adiponectin and high-density lipoprotein cholesterol concentrations [1].
Reference:
Ryo M, Nakamura T, Kihara S, Kumada M, Shibazaki S, Takahashi M, Nagai M, Matsuzawa Y, Funahashi T. Adiponectin as a biomarker of the metabolic syndrome. Circulation journal. 2004;68(11):975-81.
Response: We realize the sentence may be confusing. We are only referring to elevated levels for hs-CRP in adults with MetS.
Thus, we have amended Section 4 Discussion (page 10)
“Furthermore, we did not study cardiometabolic risk biomarkers such hs-CRP, which were shown to be elevated in adults and women post-GDM with MetS [18,21,66,67], or adiponectin, which was reduced in MetS [67,68]. Thus, the assessment of cardiometabolic risk biomarkers should be performed in future studies to better understand the physiology behind the development of MetS.”
Reviewer 4 Report
1- How did you calculate the sample size? The sample size seems to be insufficient for the cross-sectional study
2- In the material and method section you only describe the MetS but in the result section, you have Non-MetS group
3- What is the novelty of your findings?
Author Response
Comment 1: How did you calculate the sample size? The sample size seems to be insufficient for the cross-sectional study
Response: Amended 2.1 Study Design and Participants (page 3)
“Sample size was calculated using the prevalence formula for cross-sectional studies [39]. Based on the 22% prevalence of MetS in Malaysian women post-GDM [40], 95% confidence level, 10% tolerable deviation of values around the estimate and after adjusting for 20% non-response rate [39], a minimum of 83 participants was required for the study.”
As a side information, the recruitment was conducted from January to April 2021 (at Universiti Putra Malaysia), then from November 2021 to February 2022 (at Seri Kembangan Health Clinic) due to the Movement Control Order restrictions in Malaysia. Despite 8 months of data collection, recruitment was slow (1-2 postnatal patients per day). This is due to the staggered appointment of patients at the clinic, defaulted appointment, and difficulty of patients traveling inter-district (due to the restricted law) to attend the visit.
Comment 2: In the material and method section you only describe the MetS but in the result section, you have Non-MetS group
Response: Added in 2.2 Diagnosis of Metabolic Syndrome (page 3)
“Participants were then categorized into MetS (those with metabolic syndrome) or Non-MetS (those without metabolic syndrome) group.”
Comment 3: What is the novelty of your findings?
Response: Added in Section 4 Discussion (page 10)
“The study contributes data on the prevalence and prevention of MetS, in women with a previous history of GDM, which is an area with limited evidence particularly in regard to dietary factors. Western and Prudent dietary patterns in this study were also slightly different from the diets commonly practiced in Western or high-income countries. This could be due to the culturally unique food frequency questionnaire used [44], hence the dietary patterns derived in this study were more representative of the typical Asian diet.”
Round 2
Reviewer 4 Report
Thanks